# Extracellular Vesicles Derived from Human Umbilical Mesenchymal Stem Cells Transfected with miR-7704 Improved Damaged Cartilage and Reduced Matrix Metallopeptidase 13

**DOI:** 10.3390/cells14020082

**Published:** 2025-01-09

**Authors:** Kun-Chi Wu, Hui-I Yang, Yu-Hsun Chang, Raymond Yuh-Shyan Chiang, Dah-Ching Ding

**Affiliations:** 1Department of Orthopedics, Hualien Tzu Chi Hospital, Buddhist Tzu Chi Medical Foundation, Tzu Chi University, Hualien 970, Taiwan; drwukunchi@yahoo.com.tw; 2Bioinnovation Center, Buddhist Tzu Chi Medical Foundation, Hualien 970, Taiwan; s8706083@yahoo.com.tw; 3Department of Pediatrics, Hualien Tzu Chi Hospital, Buddhist Tzu Chi Medical Foundation, Tzu Chi University, Hualien 970, Taiwan; cyh0515@gmail.com; 4Department of Obstetrics and Gynecology, Hualien Tzu Chi Hospital, Buddhist Tzu Chi Medical Foundation, Tzu Chi University, Hualien 970, Taiwan; raymond880106@gmail.com; 5Institute of Medical Sciences, Tzu Chi University, Hualien 970, Taiwan

**Keywords:** extracellular vesicles, human umbilical cord mesenchymal stem cells, miRNA, MMP13, osteoarthritis

## Abstract

We aimed to explore the therapeutic efficacy of miR-7704-modified extracellular vesicles (EVs) derived from human umbilical cord mesenchymal stem cells (HUCMSCs) for osteoarthritis (OA) treatment. In vitro experiments demonstrated the successful transfection of miR-7704 into HUCMSCs and the isolation of EVs from these cells. In vivo experiments used an OA mouse model to assess the effects of the injection of miR-7704-modified EVs intra-articularly. Walking capacity (rotarod test), cartilage morphology, histological scores, and the expression of type II collagen, aggrecan, interleukin-1 beta, and matrix metalloproteinase 13 (MMP13) in the cartilage were evaluated. The EVs were characterized to confirm their suitability for therapeutic use. IL-1beta-treated chondrocytes increased type II collagen and decreased MMP13 after treatment with miR-7704-overexpressed EVs. In vivo experiments revealed that an intra-articular injection of miR-7704-overexpressed EVs significantly improved walking capacity, preserved cartilage morphology, and resulted in higher histological scores compared to in the controls. Furthermore, the decreased expression of MMP13 in the cartilage post treatment suggests a potential mechanism for the observed therapeutic effects. Therefore, miR-7704-overexpressed EVs derived from HUCMSCs showed potential as an innovative therapeutic strategy for treating OA. Further investigations should focus on optimizing dosage, understanding mechanisms, ensuring safety and efficacy, developing advanced delivery systems, and conducting early-phase clinical trials to establish the therapeutic potential of HUCMSC-derived EVs for OA management.

## 1. Introduction

Osteoarthritis (OA) is a significant global health concern, affecting approximately 528 million people worldwide in 2019 [1,2]. The incidence of OA is notably high in countries with developed countries, reaching 14% in the US [2]. In developing nations, the pooled prevalence is estimated at 16.05% [3]. China, with its large population and rapidly aging demographics, has seen the incidence of OA increase from 51.76 × 10^6^ in 1990 to 132.81 × 10^6^ in 2019 [4]. Globally, the burden of OA has steadily increased between 1990 and 2019, as reflected by the rising disability-adjusted life years to 189.49 million [1]. The economic impact of OA is significant, accounting for 1 to 2.5% of gross national product in developed market economies [2].

Current pharmacological treatments for OA primarily focus on symptom management rather than disease modification. Nonsteroidal anti-inflammatory drugs, both oral and topical, are highly recommended as first-line treatments due to their effectiveness in alleviating pain and improving function [5,6]. Intra-articular corticosteroid injections are also widely recommended with minimal adverse effects [5]. However, traditional treatments like acetaminophen are becoming less acceptable due to efficacy and safety concerns [5]. Emerging therapies include disease-modifying OA drugs that aim to reduce inflammation and promote cartilage repair [6]. Nanotechnology-based approaches, such as liposomes, micelles, and polymeric nanoparticles, show promise for OA treatment [7]. Small-molecule inhibitors targeting cartilage remodeling are also being explored as potential new treatment options [8]. Stem cell therapy, especially that utilizing mesenchymal stem cells (MSCs), shows promise for cartilage regeneration and alleviating inflammatory response associated with OA [9,10].

Human umbilical cord mesenchymal stem cells (HUCMSCs) represent a potential new stem cell source with advantages such as non-invasiveness and fasters self-renewal compared to the bone marrow [11,12]. HUCMSCs possess tri-lineage differentiation potential and immunomodulatory capabilities, making them attractive candidates for use in autologous or allogeneic transplantation to treat degenerative diseases, such as OA [13,14]. However, challenges remain in MSC therapy, including nonspecific therapeutic targeting and the intricacy of cell culture and injection procedures [11].

Therefore, extracellular vesicles (EVs), small vesicles secreted from cells, have evolved as promising alternatives for OA therapy [15]. EVs are heterogeneous membrane-enclosed structures ranging from 20 to 5000 nm in size, classified into exosomes, microvesicles, and apoptotic vesicles [16,17,18]. EVs contain proteins and ribonucleic acid (RNAs), including miRNAs, which play crucial roles in regulating gene expression and modulating the inflammatory response in OA joints [12,19]. MSC-derived extracellular vesicles offer several advantages over direct cell injection, including the ease of administration and a reduced immune response [20]. Specifically, MSC-derived extracellular vesicles decrease inflammation, promote cartilage matrix synthesis, and reduce cartilage degradation in preclinical studies [21,22].

MicroRNAs (miRNAs), a category of RNA that does not encode proteins, have been involved in OA pathology and can regulate key signaling pathways of cartilage homeostasis and degradation [23]. Several miRNAs, including miR-9, miR-27, miR-34a, miR-140, and miR-146a, have been identified as dysregulated in OA, affecting inflammatory pathways and extracellular matrix degradation [24]. MiR-140 directly targets ADAMTS-5, an aggrecanase, while miR-146a inhibits MMP13 and ADAMTS4, both cartilage-degrading enzymes [25]. Bioinformatic analyses have revealed 46 differentially expressed miRNAs involved in various aspects of OA, such as autophagy, inflammation, and chondrocyte metabolism [26]. These findings suggest that miRNAs may serve as potential diagnostic biomarkers and therapeutic targets for OA [24,26]. As research progresses, new miRNA targets are being validated, opening possibilities for novel therapies to control joint destruction and stimulate cartilage repair [25].

miR-7704 has been identified as a potential therapeutic target for acute lung injury, promoting M2 macrophage polarization and improving pulmonary function [27]. We previously found that miR-7704 was enriched in the condition medium of HUCMSCs. However, the therapeutic role of miR-7704 on osteoarthritis is not known. 

Matrix metallopeptidase 13 (MMP13) is a key enzyme in cartilage breakdown [28]. MMP13 plays a crucial role in OA progression by degrading type II collagen in articular cartilage [28,29]. Studies have shown that MMP13 inhibition can decelerate OA progression and protect cartilage integrity in animal models [29]. Serum MMP13 levels are associated with various knee structural abnormalities, including reduced cartilage volume, increased cartilage defects, and altered infrapatellar fat pad characteristics [30]. The regulation of MMP13 in OA involves complex networks, including epigenetic factors, non-coding RNAs, and autophagy-related proteins [31]. Additionally, inflammatory factors such as TNF-α, IL-8, and IL-18 are positively associated with serum MMP13 levels, while adiponectin shows a negative association [30]. These findings highlight MMP13 as a promising target for early OA diagnosis, monitoring, and treatment development [28,31].

miR-410 promotes the chondrogenic differentiation of MSCs by targeting Wnt3a, inhibiting the Wnt signaling pathway [32]. Similarly, miR-335 expression is altered in OA-derived MSCs, potentially affecting Wnt signaling and OA progression [33]. The chondrogenic induction of OA-derived MSCs activates mineralization and hypertrophic gene expression, regulated by miR-365, a mechanosensitive microRNA [34]. These findings highlight the potential of miRNAs as therapeutic targets for OA treatment. MSC transplantation is a promising approach for OA therapy due to their paracrine effects, anti-inflammatory activity, and differentiation potential. However, strategies to induce specific differentiation, such as using miRNAs, are needed to overcome challenges associated with transplanting naïve MSCs [35].

Previous studies have explored other miRNAs (miR-10a-5p, miR-210) for their therapeutic potential [36,37], but miR-7704′s role in OA and its efficacy in promoting cartilage repair through HUCMSC-derived extracellular vesicles are relatively unknown. Therefore, this study addresses this gap by overexpressing miR-7704 in HUCMSCs and evaluating their EVs’ treatment effects in a mouse OA model. By elucidating the underlying mechanisms of miRNA transport by EVs and its effects on cartilage preservation and inflammation reduction, we sought to provide insights into a novel and promising approach for OA management.

## 2. Materials and Methods

### 2.1. Human Umbilical Mesenchymal Stem Cell Culture

We used HUCMSCs in our study due to our previous extensive experience with HUCMSC derivation and culture. The experiment protocol was approved by the Research Ethics Committee of the Buddhist Tzu Chi General Hospital (IRB 111-230-B). All participants provided their written informed consent.

We used the protocol for deriving HUCMSCs as our previously published materials [19]. Briefly, we cut a 20 cm long human umbilical cord after baby delivery and put it in a sterile tube filled with Hanks’ balanced salt solution (HBSS; Gibco/BRL 14185-052, Grand Island, NY, USA). The cord would be processed within 24 h. The cord was washed with calcium- and magnesium-free phosphate-buffered saline (PBS; Biowest, Nuaille, France) three times, and then the umbilical artery, vein, and outer membranes from Wharton’s jelly were cut out. The jelly was cut into small pieces, followed by the digestion of type 1 collagenase (Sigma, St. Louis, MO, USA). The resulting materials were placed in the incubator with a 37 °C humidity environment and a 95% air/5% CO_2_ atmosphere for 14–18 h. The culture media were composed of low-glucose Dulbecco’s modified Eagle’s medium (DMEM; Gibco, Grand Island, NY, USA) and 10% fetal bovine serum (FBS; Biological Industries, Kibbutz, Israel). We left the culture undisturbed for 5–7 days to enable cell proliferation from the small pieces of jelly. The proliferated cells were designated as the first generation of HUCMSCs. Once they reached 80% confluence, typically after 5–7 days, the cells were subcultured. Following trypsinization, the cells were split at a 1:3 ratio for passage.

### 2.2. Flow Cytometry

HUCMSCs were mixed with fluorescent dye-conjugated antibodies and placed on ice for 30 min. After incubation, the cells were washed twice with PBS to remove unbound antibodies. The cells were incubated with a blocking solution (PBS containing 2% FBS) for an additional 10–15 min to block nonspecific binding. Finally, the stained cells were analyzed using flow cytometry or fluorescence-activated cell sorting (FACSCalibur; BD Biosciences, Franklin, NJ, USA) to assess the expression of the target markers. We used unstained cells to account for background fluorescence as negative controls. The antibodies used were CD29, CD34, CD44, CD45, CD73, CD105, HLA-DR, and HLA-ABC (all purchased from BD Biosciences).

### 2.3. Cell Proliferation

Cell proliferation was assessed using the XTT (2,3-Bis-(2-Methoxy-4-Nitro-5-Sulfophenyl) kit, Biological Industries) assay. A total of 2 × 10^3^ cells were seeded in each well of a 96-well plate, with 100 μL of culture medium added to each well. Following this, 150 μL of XTT solution was added, and the cells were cultured for 3 h at 37 °C. After incubation, the plate was transferred to a microplate reader (Model 3550, Bio-Rad, Hercules, CA, USA), and absorbance was measured at 450 nm on days 0, 3, and 7. The optical density values obtained were plotted to create a proliferation curve.

### 2.4. Trilineage Differentiation Capabilities

#### 2.4.1. Adipogenesis

To induce adipogenesis in stem cells, 5 × 10^4^ cells were seeded per well in a 12-well plate and allowed to reach 60–80% confluence. Once the cells were ready, we replaced the growth medium with adipogenic differentiation medium. The adipogenic medium was composed of DMEM supplemented with 10% FBS, 5 μg/mL insulin, 1 μmol/L dexamethasone, 60 μmol/L indomethacin, and 0.5 mmol/L isobutylmethylxanthine (all purchased from Sigma). We cultured the cells in this induction medium for a period of 14 days and changed the medium every 2 to 3 days. During this time, the stem cells began to differentiate into adipocytes, accumulating lipid droplets in the cytoplasm. After 14 days, we confirmed adipogenesis by staining the cells with Oil Red O (Sigma), a dye that specifically stains lipid droplets. Lipid concentration was measured by the absorbance at 510 nm in a microplate reader. All experiments were carried out thrice.

#### 2.4.2. Osteogenesis

To induce osteogenesis in stem cells, 1 × 10^4^ cells were seeded per well in a 12-well plate and allowed to reach 60–80% confluence. Once the cells were ready, we replaced the growth medium with an osteogenic differentiation medium. The induction medium was composed of DMEM supplemented with 10% FBS, 50 μmol/L ascorbate, 0.1 μmol/L dexamethasone, and 10 mmol/L β-glycerol phosphate (all purchased from Sigma). We cultured the cells in this osteogenic medium for 14 days and changed the medium every 2 to 3 days to maintain the induction conditions. During this period, the stem cells would begin to differentiate into osteoblast-like cells, generating mineralized deposits within the extracellular matrix. After 14 days, osteogenesis was confirmed by staining the cells with Alizarin Red S (Sigma), which stains calcium deposits and indicates the formation of mineralized bone-like structures. We quantified staining by adding 800 μL 10% (*v*/*v*) acetic acid for 30 min to solve stain materials. Then, the resulting materials were read at 562 nm using a 96-well plate with opaque walls and transparent bottoms (Fisher Scientific, Hampton, NH, USA) with an ELISA reader.

#### 2.4.3. Chondrogenesis

To induce chondrogenesis in stem cells using the pellet method, we centrifuged 2.5 × 10^7^ cells/mL to obtain a cell pellet and resuspended the cells in a chondrogenic differentiation medium. The medium was composed of DMEM supplemented with 10% FBS, 6.25 μg/mL insulin (Sigma), 10 ng/mL transforming growth factor-β1 (Pepro Tech, Rocky Hill, NJ, USA), and 50 μg/mL ascorbic acid-2-phosphate (Sigma). We seeded approximately 2.5 × 10^7^ cells per 15 mL of conical tube and allowed the cells to aggregate and form a pellet by centrifuging at a low speed (around 1000 rpm) for 5 min. The pellet was subsequently maintained in a chondrogenic medium for 21 days, with the regular medium changed every 2–3 days to support differentiation. During this period, the stem cells would differentiate into chondrocytes and produce extracellular matrix components, including proteoglycans and collagen. After 21 days, chondrogenesis was confirmed by staining type II collagen and aggrecan to detect glycosaminoglycans and proteoglycans, key indicators of cartilage formation.

### 2.5. Immunohistochemical Staining

To perform the immunohistochemistry staining of the pellet for aggrecan and type II collagen, we fixed the pellet in 4% paraformaldehyde for 24 h at 4 °C to preserve tissue morphology. After fixation, the pellet was embedded in an optimal cutting temperature (OCT) compound and cryosectioned into 10–20 µm thick sections. Once the sections were prepared, we incubated them at room temperature for 30 min to allow for optimal adhesion to the slides. We blocked nonspecific binding by incubating the sections in a blocking solution (5% normal serum and 0.1% Triton X-100 in PBS) for 1 h at room temperature. After blocking, we incubated the sections overnight at 4 °C with primary antibodies specific for type II collagen and aggrecan (1:200; Sigma-Aldrich, St. Louis, MO, USA). Following primary antibody incubation, we washed the sections three times with PBS and then incubated them with horseradish peroxidase (HRP) for 1 h at room temperature. After washing, the stained sections were visualized using a fluorescence microscope (for fluorescent detection) or a light microscope (for HRP detection, following DAB substrate application; all purchased from Sigma). Finally, we counterstained the sections with hematoxylin for nuclei visualization and mounted them with a mounting medium for analysis. The expression of type II collagen and aggrecan would indicate the successful chondrogenesis of the stem cells in the pellet.

### 2.6. Quantitative Polymerase Chain Reaction

Total RNA was isolated from cells using the PureLink RNA Mini Kit (Life Technologies, Carlsbad, CA, USA). The SuperScript III enzyme (Invitrogen, Waltham, MA, USA) was used to reverse-transcribe 1 μg of total RNA into complementary DNA (cDNA). Quantitative real-time qPCR was carried out with the Fast SYBR Green Master Mix (Applied Biosystems, Foster City, CA, USA), and data analysis was conducted using Quant Studio 5 (Applied Biosystems, Waltham, MA, USA). We used glyceraldehyde-3-phosphate dehydrogenase (*GAPDH*) as the internal control. All experiments were performed in triplicate. The genes of interest included markers of trilineage differentiation: *PPARγ* and *FABP4* for adipogenesis, *ALPL* and *RUNX2* for osteogenesis, and *Aggrecan* and *COL2A1* for chondrogenesis. PCR using water and non-reverse-transcribed mRNA was regarded as a control. Melting curve analysis and agarose gel electrophoresis were used to confirm the PCR product specificity. QPCR software (7500 Software v2.3, Applied Biosystems) was used to determine each gene’s threshold cycle (Ct) values. The results were normalized to the GAPDH expression using the 2^−ΔΔCt^ method [38]. Table 1 lists the primer sequences.

### 2.7. Transfection of miR-7704

miR-7704 was encoded into lentivectors and transfected into HUCMSCs. A lentiviral vector carrying the miR-7704 sequence (CGGGGUCGGCGGCGACGUG) was prepared in the laboratory; this involved cloning the miR-7704 sequence into a lentiviral expression vector along with the necessary regulatory elements in the promoter to drive its expression. Thereafter, the miR-7704 sequence–carry lentiviral vector was transfected into packaging cells along with other necessary viral components obtained from ABM Company (LentimiRa-GFP-hsa-mir-7704 Virus, cat. no. mh16481, New York, NY, USA). These cells produced lentiviral particles that incorporated the miR-7704 sequence. Lentiviral particles containing the miR-7704 sequence were harvested from the culture supernatant of the packaging cells. This involved collecting viral particles, which were concentrated and purified to obtain a high-titer lentiviral stock. One day before transfection, 1 × 10^5^ cells were seeded in a 6-well plate with 2 mL of DMEM/F12 containing 10% FBS and 1X penicillin/streptomycin. HUCMSCs were exposed to lentiviral particles containing the miR-7704 sequence by adding the lentiviral stock directly to the cell culture medium (MOI = 2) for 48 h. For 1 week, the transduced cells were selected using antibiotic selection (puromycin 0.8 μg/mL). Successfully transduced cells were examined using fluorescence microscopy to detect green fluorescent protein (GFP) expression before expansion.

### 2.8. Extracellular Vesicle Extraction and Characterization

EVs were extracted from the conditioned medium (CM) of HUCMSCs, +lenti-control, and +miR-7704 cultured for 48 h. When the cells reached 50–60% confluence, they were washed with PBS and cultured in a serum-free defined medium (MesenGRo hMSC medium, StemRD, Burlingame, CA, USA) for an additional 48 h. The CM was collected and subjected to centrifugation at 300× *g* for 15 min, followed by a second centrifugation at 2500× *g* for another 15 min to eliminate cell debris and dead cells. After centrifugation, the supernatant was passed through a 0.22 μm filter (Merck–Millipore, Burlington, MA, USA) to remove any remaining cells and debris. ExoQuick (63 µL, EXOTC50A-1, System Biosciences, Palo Alto, CA, USA) was added to the sample (250 µL), and incubated for at least 1 h at 4 °C for EV isolation. After a low-speed centrifuge (2000× *g*) for 30 min, the supernatant was removed. The remaining EV pallet was centrifuged at 1500× *g* for 5 min to eliminate any residual ExoQuick solution. The ExoQuick isolation process was sterile. The isolated EVs were stored in EV-Guard EV storage buffer (EXSBA-1, System Biosciences, Palo Alto, CA, USA) and kept frozen at −80 °C.

### 2.9. Extracellular Vesicle Identification

Extracellular vesicles were identified via Western blotting for general extracellular vesicle markers (CD63, CD9, and CD81), NanoSight analysis, and transmission electron microscopy (TEM).

### 2.10. Western Blot Analysis

Western blotting was performed to assess the expression levels of CD9, CD63, and CD81. Extracellular vesicles were broken down with a protein lysis buffer (Sigma-Aldrich) to obtain the proteins. The proteins were then separated by 10% sodium dodecyl sulfate-polyacrylamide gel electrophoresis (SDS-PAGE; Sigma-Aldrich). Membranes were incubated overnight at 4 °C with primary antibodies against CD9, CD63, and CD81 (all from Sigma-Aldrich), diluted at 1:2000. Afterward, the membranes were incubated with a secondary antibody (horseradish peroxidase (HRP); Sigma), diluted at 1:5000. An electrochemiluminescence kit (Promega, Fitchburg, WI, USA) was used to detect HRP signals. For in vitro experiments, the chondrocytes were treated with IL-1β (10 ng/mL) for 24 h, followed by culture with or without MSC-EVs, lenti-control EVs, or 7704-EVs. Then, the protein of the four groups was extracted, and the expressions of aggrecan, type II collagen, and MMP-13 (all from Sigma-Aldrich) were analyzed.

### 2.11. NTA (NanoSight NS300)

EVs were analyzed using a Nano-Sight NS300 instrument (Malvern, Worcestershire, UK). Samples were diluted with 0.2 μm filtered 1× normal saline to achieve an optimal particle concentration range of 20–150 particles per frame, typically by diluting 100× or 1000×. We recorded three 60 s videos for each sample and used NanoSight NTA Software 3.2 (Malvern, Worcestershire, UK) to analyze the data. The post-acquisition settings were standardized and applied consistently to all samples. The resulting data provided information on particle concentration (particles/mL), particle size, and the distribution of average particle size (nm).

### 2.12. TEM

TEM was used to observe the extracellular vesicles by providing high-resolution images of their ultrastructure after isolation and purification, followed by fixation with 2% glutaraldehyde (Sigma) in phosphate buffer (pH 7.4, Sigma) to preserve their structure. A small drop (3 µL) of the fixed extracellular vesicle suspension was applied to a carbon-coated TEM grid (697745, Sigma) and then washed and stained with a contrasting agent: 1% uranyl acetate (21447-25, Polysciences, Warrington, PA, USA). After air-drying, the grid was placed in the TEM (HITACHI H-7500, Tokyo, Japan), where an electron beam generated detailed images of the extracellular vesicles, typically appearing as round, cup-shaped vesicles with a 30–150 nm diameter. The exosomal morphology, size, and structural features were recorded.

### 2.13. Chondrocyte Culture

Human chondrocytes were extracted from cartilage samples from humans. We cut cartilage into small pieces and digested it with type II collagenase diluted in a culture medium. The chondrocyte culture medium consisted of DMEM/F12 (Sigma) supplemented with 10% FBS and 1% penicillin/streptomycin. After overnight digestion, the mixture was filtered through a 70 μm cell strainer (Falcon, BD Biosciences, Franklin Lakes, NJ, USA) to remove debris, and the cells were plated in a 10 cm dish with 10 mL of culture medium. The chondrocytes were cultured for 14 days, after which they were passaged.

### 2.14. Osteoarthritis Mouse Model Induced by Type II Collagen

We obtained approval for the animal experiments from the Institutional Animal Care and Use Committee of Buddhist Tzu Chi General Hospital. The animal experiments were performed according to the relevant guidelines and regulations.

Due to the inflammatory and immune-mediated nature of the OA environment, immunocompetent mice were used in this study. For in vivo EV experiments, female B6 mice (8 weeks old) were employed as the animal model to assess the effects of cartilage repair. The mice were divided into five groups. The first group was the normal control group receiving only normal saline (*n* = 3) without any treatment. Seven days after collagenase injection, mice displaying significant movement disabilities were selected for further studies. The 2nd group was the mice treated with collagenase (C0773, Sigma-Aldrich, St. Louis, MO, USA) followed by normal saline (OA group, *n* = 3). The other three groups were composed of mice treated with collagenase (Sigma) and receiving EVs derived from HUCMSCs (MSC-EVs, lenti-control EVs, and miR7704-EVs; *n* = 3 per group).

### 2.15. Extracellular Vesicle Injection

In mice after collagenase injection for 7 days, three treatment groups were injected with EVs (concentration: 1 × 10^7^ particles/mL) in 50 μL of PBS, the procedure was conducted under anesthesia using ketamine (50 mg/kg) and xylazine (15 mg/kg). The injection points were positioned beneath the infrapatellar ligament of both hind leg knees. After the transplantation of the EVs, the mice were permitted to move around freely and eat as they wished in their cages.

### 2.16. Behavioral Assessments

All mice underwent 3-day training on the rotarod system (3376-4R, TSE Systems, Chesterfield, MO, USA) to familiarize themselves with the equipment. The average duration of staying time was recorded for each mouse. Only mice that successfully completed the training were included in the study. After OA induction, testing was conducted at 7-day intervals during the light phase. Mice were placed on the rotarod, and their staying time during ambulation was recorded. Behavioral tests were performed across all groups on days 0 (before collagenase injection), 7 (following normal saline or EVs injections), 14, 21, and 28. Each time point consisted of five testing trials, and the mean staying time was calculated. The maximum test duration was set at 120 s, with the rotarod speed maintained at 20 rpm.

### 2.17. Tissue Harvesting

After conducting the experiment for 28 days, the mice were euthanized using CO_2_ and decapitation. We dissected the hind knee joints on both sides and analyzed the joint surfaces. The distal femur and proximal tibial plateaus were collected and preserved in 10% buffered formalin (Sigma) for 48 h. The samples were then decalcified in 10% ethylenediaminetetraacetic acid (EDTA, Gibco, Waltham, MA, USA) for two weeks. Once decalcification was complete, the specimens were cut into four pieces, embedded in paraffin, and prepared as serial sagittal sections. These sections were stained with hematoxylin and eosin (H&E, Sigma) and Safranin O (Sigma) to assess histological changes, which were then examined under a microscope.

### 2.18. Histological Evaluation

A histological evaluation of the samples was conducted using the International Cartilage Repair Society (ICRS) scoring system [39]. Two evaluators performed the assessments independently and in a blind manner. The ICRS scoring system comprises six categories: cell distribution, surface characteristics, matrix composition, cell viability, cartilage mineralization, and subchondral bone integrity. Scores range from 0 to 18, with higher scores indicating superior cartilage morphology.

### 2.19. Immunohistochemical Staining

Immunohistochemistry was conducted to assess the expression levels of type II collagen, aggrecan, IL-1β, and MMP-13 in tissue samples. The paraffin-embedded sections were first deparaffinized using xylene and then rehydrated through a series of graded ethanol solutions. Antigen retrieval involved heating the sections in citrate buffer (pH 6.0) at 95 °C for 20 min. Once cooled to room temperature, the sections were treated with 3% hydrogen peroxide to eliminate endogenous peroxidase activity, followed by incubation with 5% bovine serum albumin (BSA) to prevent nonspecific binding. The sections were then incubated overnight at 4 °C with primary antibodies targeting type II collagen, aggrecan, IL-1β, and MMP-13 (1:100, GeneTex, Irvine, CA, USA). After washing with PBS, the sections received an HRP-conjugated secondary antibody (GeneTex) for 1 h at room temperature. Signal detection was carried out using a DAB substrate kit, and the sections were counterstained with hematoxylin. The stained sections were analyzed under a light microscope (Nikon TE2000-U, Tokyo, Japan) for both qualitative and quantitative assessment, with the percentage of positively stained cells calculated by examining five randomly chosen fields.

### 2.20. Statistical Analysis

The results were presented as means with standard errors of the mean (SEM). The data from qRT-PCR, rotarod tests, and histological assessments were analyzed using one-way repeated measure ANOVA and ANOVA followed by Fisher’s least significant difference (LSD) post hoc test. To compare the rotarod performance between groups at the same time point, two-way ANOVA was utilized. A *p*-value of less than 0.05 was deemed statistically significant.

## 3. Results

### 3.1. HUCMSCs Revealed Typical Characteristics of MSCs

Figure 1 characterizes the properties and differentiation potential of human umbilical cord mesenchymal stem cells (HUCMSCs). Figure 1A shows the flow cytometry results, confirming that HUCMSCs expressed high levels of mesenchymal markers (CD44, CD73, CD90, CD105, and HLA-ABC) while lacking expression of hematopoietic markers (CD34, CD45) and HLA-DR. Figure 1B,C illustrate successful adipogenic differentiation, with Oil Red O staining indicating lipid accumulation in the differentiated cells, quantified by a significant increase in optical density (O.D. 510). Figure 1E,F demonstrate osteogenic differentiation, evidenced by Alizarin Red staining for calcium deposits, with quantification showing a significant increase in optical density (O.D. 562). Figure 1D, 1G, and 1I present the upregulation of key adipogenic (*FABP4*, *PPARγ*), osteogenic (*ALPL*, *RUNX2*), and chondrogenic (*ACAN*, *COL2A1*) markers, respectively, in lineage-specific differentiation media. Figure 1H shows histological staining, where H&E, aggrecan, and type II collagen staining confirmed successful chondrogenic differentiation, further validating the multipotency of HUCMSCs.

### 3.2. miR-7704 Successfully Transfected into HUCMSCs

Figure 2 illustrates the characterization of miR-7704 expression in cells and EVs. Figure 2A shows fluorescence microscopy images of cells transfected with miR-7704-GFP, displaying strong green fluorescence (left), corresponding phase-contrast morphology (middle), and merged images (right), confirming successful transfection and GFP expression. Figure 2B presents a bar graph comparing the relative miR-7704 levels in EVs derived from 1121WJ cells (HUCMSCs), Lenti-Ctrl exosomes, and miR-7704 exosomes. miR-7704 EVs demonstrated significantly higher levels of miR-7704 compared to both 1121WJ and Lenti-Ctrl EVs (*p* < 0.001), validating the effective overexpression of miR-7704 in EVs.

Figure 2C presents NTA showing the size distribution and concentration of exosomes. HUCMCS-e exosomes had an average size of 124.5 nm with a concentration of 1.2 × 10^9^ particles/mL, Lenti-e exosomes had a mean size of 111.7 nm with the same concentration, and miR-7704-e exosomes were slightly smaller, averaging 99.5 nm with a higher concentration of 3.1 × 10^9^ particles/mL. Figure 2D shows TEM images confirming the spherical morphology of exosomes for all three groups, with sizes consistent with the NTA results, further validating the successful isolation of intact exosomes.

Figure 2E presents a Western blot analysis of the tetraspanin proteins CD9 (~20 kDa), CD63 (~60 kDa), and CD81 (~20 kDa), which are established markers of EVs. Three samples—HUCMSC-e, Lenti-e, and miR-7704-e—were analyzed. CD9 expression was observed in all samples but was more prominent in the Lenti-e and miR-7704-e groups compared to HUCMSC-e. CD63 showed consistent expression across all samples without significant differences in intensity, while CD81 was similarly expressed in all three conditions. These results confirm the presence of EVs markers in the samples, with notable variation in CD9 levels. Taken together, miR-7704 was successfully transfected into HUCMSCs and EVs derived from various cells, fulfilling the EV criteria.

### 3.3. Impact of HUCMSC-miR-7704-e on Type II Collagen and MMP13 Protein Expression Levels in the IL-1β-Treated Chondrocytes

IL-1β plays a crucial role in osteoarthritis and fracture healing by modulating chondrocyte behavior and cartilage degradation. IL-1β stimulates the expression of MMP-13 and decreases type II collagen (COL2A1) and aggrecan production in chondrocytes [40,41,42]. We induced cartilage degradation using IL-1β and evaluated how EVs regulate these proteins.

Figure 3 presents data on the effects of IL-1β stimulation and various treatments using HUCMSCs on the expression levels of aggrecan, COL2A1, and MMP13 proteins. Figure 3A displays Western blot images showing that IL-1β alone reduced the expression of aggrecan and COL2A1, while increasing MMP13 levels. The treatment with HUCMSC-miR-7704-e notably reversed these effects, showing decreased MMP13 and increased COL2A1 protein levels, while aggrecan expression remained relatively stable across treatments. Figure 3B–D present quantifications of the protein expression for MMP13, COL2A1, and aggrecan, respectively. In particular, MMP13 levels (B) were significantly reduced with the HUCMSC-miR-7704-e treatment compared to the other groups, while the COL2A1 levels (C) were significantly elevated in the HUCMSC-miR-7704-e group. Aggrecan levels (D) showed no significant changes across treatment groups. GAPDH was used as a loading control to ensure equal protein loading.

These findings reveal the potential therapeutic role of HUCMSC-miR-7704-e in modulating MMP13 and COL2A1 expression on IL-1β-treated chondrocytes.

### 3.4. Rotarod Behavior Improved After EV Treatments

Figure 4A evaluates the effects of various exosome treatments on Rotarod performance in an OA model. Figure 4A demonstrates that EV treatments, particularly lenti-miR7704-e, significantly improved Rotarod performance (latency to fall) over time compared to the OA group, indicating enhanced motor function.

### 3.5. Histology of Cartilage of Knee Joint

Figure 4B presents histological images of joint tissues stained with H&E and Safranin O, showing improved cartilage integrity and proteoglycan retention in groups treated with HUCMSC-e, lenti-ctrl-e, and lenti-miR7704-e exosomes compared to the OA group, with the lenti-miR7704-e group showing the most notable improvement. Figure 4C quantifies the ICRS histological scores, highlighting that the lenti-miR7704-e group achieved a significant increase compared to the OA group, similarly to the +SF and lenti-ctrl exosome groups, though only the lenti-miR7704-e group showed statistically significant results. 

### 3.6. Comparable Expression of Chondrogenic Proteins Post-Transplantation of EVs

Figure 5 investigates the effects of exosome treatments on the expression of Type II collagen and aggrecan in an OA model. Figure 5A shows the immunohistochemical staining for type II collagen, with the OA group exhibiting reduced staining compared to the normal group. Treatment with HUCMSC-e, lenti-ctrl-e, and lenti-miR7704-e exosomes restored type II collagen expression, with the lenti-miR7704-e group displaying the most pronounced improvement. Figure 5B quantifies the percentage of cells positive for type II collagen, indicating an increase in EV-treated groups compared to the OA group, though the differences were not statistically significant. Figure 5C shows aggrecan staining, where EV-treated groups preserved aggrecan expression similar to the normal group, while the OA group exhibited reduced staining. Figure 5D quantifies the percentage of cells positive for aggrecan, revealing no significant differences among the groups, suggesting that EV treatments maintained the cartilage matrix protein expression.

### 3.7. Decreased MMP13 in EV-Treated Cartilages

Figure 6 evaluates the effects of exosome treatments on IL-1β and MMP13 expression in an OA model. Figure 6A shows immunohistochemical staining for IL-1β, with the OA group exhibiting increased staining compared to the normal group. Exosome treatments, including HUCMSC-e, lenti-ctrl-e, and lenti-miR7704-e, appeared to reduce IL-1β expression, although Figure 6B quantifies the percentage of IL-1β-positive cells, showing no statistically significant differences among the groups. Figure 6C presents the MMP13 staining, revealing markedly elevated expression in the OA group compared to the normal group. EV treatments, particularly lenti-miR7704-e, visibly reduced MMP13 expression. Figure 6D quantifies the percentage of MMP13-positive cells, showing that lenti-miR7704-e significantly decreased MMP13 expression compared to the OA group, aligning with its potential therapeutic effect. Statistically significant differences were also observed between the OA group and other exosome treatments, underscoring the efficacy of lenti-miR7704-e.

## 4. Discussion

This study thoroughly explains the mechanisms underlying the therapeutic effects of HUCMSC-derived extracellular vesicles, including the modulation of MMP13 expression in the cartilage. In addition, this study adds to existing knowledge by highlighting the potential of HUCMSC-derived extracellular vesicles, particularly those transfected with miR-7704, as a potential therapeutic approach for OA.

EVs are essential in OA pathogenesis and treatment. EVs also show therapeutic potential in OA treatment. Mesenchymal stem cell-derived EVs have demonstrated anti-inflammatory effects and pain relief in OA [43]. Studies have shown promising results in cartilage repair and the treatment of OA using EV therapies in animal models [43,44]. EVs can regulate the OA microenvironment by delivering bioactive molecules, including nucleic acids and proteins [45]. Although human clinical trials are yet to be conducted, EVs offer the potential for developing novel OA treatments and diagnostic biomarkers [43,46].

Figure 4 demonstrates that EV injection significantly improved Rotarod performance and the ICRS histological score, suggesting that this effect is independent of transfecting HUCMSCs with lenti-miR-7704. We speculate that EVs derived from HUCMSCs might contain sufficient levels of miR-7704, which could contribute to the observed outcomes. Regarding the significant miR-7704 expression observed in lenti-control EVs, the vector primarily contained mCherry, unlike the lenti-miR-7704 vector. However, it remains unclear why the lenti-control vector also led to increased miR-7704 expression compared to the negative control.

EVs have emerged as promising carriers for miRNA delivery in OA treatment. EVs can transport miRNAs across joint tissues, influencing OA pathogenesis and cartilage homeostasis [47,48]. Specifically, miR-140 delivery to chondrocytes via engineered exosomes has shown potential in alleviating OA progression in animal models [49]. EVs have the ability to modulate the OA microenvironment by modulating biochemical factors and participating in intercellular communication [45]. However, challenges remain in the precise quantitative management of EV-miRNA therapies, which may hinder clinical translation [48]. Despite these obstacles, EV-mediated miRNA delivery represents a promising approach for OA treatment, offering potential advantages over traditional therapies by targeting the underlying disease mechanisms rather than merely addressing symptoms [45].

MSC-derived EVs can transfer miRNAs to target cells, mimicking the therapeutic effects of MSCs themselves [50]. Specific miRNAs, such as miR-150-3p and miR-223, have shown promise in maintaining chondrocyte function and suppressing OA progression [51,52]. Wang et al. demonstrated EV-mediated miR-150-3p delivery from fibroblast-like synoviocytes to chondrocytes by regulating the innate immune response to maintain joint homeostasis [51]. Liu et al. engineered EVs with enhanced cartilage-targeting capabilities and miR-223 loading, showing improved OA treatment efficacy [52]. These findings suggest that EV-mediated miRNA delivery represents a novel therapeutic strategy for OA, potentially serving as both a treatment and a biomarker for disease progression. Our results align with those of previous studies that investigated the therapeutic effects of the MSC-derived extracellular vesicle overexpression of miR-7704 in OA.

miRNAs are important in OA pathogenesis by controlling inflammatory pathways and MMP-13-mediated cartilage degradation [24]. EVs have emerged as carriers for miRNAs, facilitating their exchange within cartilage and between joint tissues [47]. Previous studies have focused on the therapeutic ability of EV-mediated miRNA transfer in OA treatment [48]. Notably, miR-140 has shown promise in inhibiting cartilage-breaking protease and alleviating the progression of OA. Researchers have developed chondrocyte-targeting exosomes by combining a chondrocyte-affinity peptide with exosome surface proteins, achieving the efficient delivery of miR-140 to deep cartilage regions [49]. This approach demonstrates the potential for targeted miRNA delivery in OA therapy. However, challenges remain in the precise quantitative management of EV-miRNAs, which may impact their clinical translation [24,53].

EVs are important for intercellular communication and disease biomarkers, but their isolation remains challenging. Various methods for EV purification from biological fluids have been compared, including ultracentrifugation, commercial kits like ExoQuick^TM^, and ultrafiltration [54,55]. These studies found that different techniques yield varying EV concentrations, sizes, and protein content. Commercial precipitation reagents like ExoQuick^TM^ generally showed higher efficiency in EV enrichment compared to traditional ultracentrifugation [55,56]. However, protocol modifications may be necessary for specific applications; for instance, thrombin should be omitted when isolating EVs from blood to prevent entrapment in clots [57]. The choice of isolation method can significantly impact downstream analyses and potential clinical applications. Therefore, developing standardized, reproducible protocols for EV isolation is crucial for advancing research and therapeutic applications in this field [54].

EVs are promising for therapeutic applications, but their storage and preservation pose challenges. Studies indicate that −80 °C is the most favorable storage condition for both biofluids and isolated EVs [58,59]. However, storage at −80 °C can still lead to a reduction in EV concentration, changes in size and zeta potential, and decreased sample purity over time [60]. Freeze–thaw cycles can further impact EV properties, with the first cycle causing significant particle loss [60]. Storage conditions affect not only physical properties but also the functional aspects of EVs, such as their antibacterial effects [61]. While −80 °C storage allows for the partial preservation of function for up to 28 days, freshly prepared EVs are recommended for functional tests [61]. These findings highlight the importance of considering storage conditions in EV research and potential clinical applications.

Quality control assays are crucial for ensuring EV safety and efficacy, yet remain poorly designed [62]. Regulatory compliance requires categorizing EVs as either active drug components or delivery vehicles, influencing manufacturing and clinical investigation requirements [63]. Large-scale EV production involves GMP-based processes, bioengineering, and quality assessments before human trials [64]. Novel isolation approaches, characterization techniques, and manufacturing considerations are essential for consistent and scalable EV production [65]. Adherence to Good Manufacturing Practice (GMP) guidelines and Minimal Information for Studies of Extracellular Vesicles (MISEV) standards is critical for standardized production processes [65]. Despite growing interest in EV-related clinical programs, obtaining marketing authorization remains complex due to the lack of specific regulatory guidelines for these novel products [64].

EVs show promise as therapeutic agents and drug delivery systems due to their biocompatibility, targeting capabilities, and biological activities [66,67,68]. However, clinical translation faces several challenges. These include low yield, complicated isolation procedures, and low loading efficiency [69]. To overcome these hurdles, researchers are developing strategies for high-yield production, efficient cargo loading, and optimized manufacturing processes [69]. The standardization of EV characterization and quality control is crucial for clinical implementation [70]. Additionally, large-scale production methods compliant with GMP are needed [66]. Despite these challenges, EVs offer potential advantages over synthetic nanocarriers in terms of targeting, safety, and pharmacokinetics [71]. As the field progresses, EVs may become a valuable addition to the therapeutic arsenal for treating various human pathologies, including degenerative, metabolic, and cancerous diseases [71].

The strengths of this study include the following: First, we employed a well-designed mouse model to assess HUCMSC-derived EVs’ therapeutic efficacy in OA treatment, utilizing various histological and immunohistochemical analyses. Second, by investigating MMP13 modulation in the cartilage, this study offers a mechanistic understanding of the therapeutic effects of HUCMSC-derived extracellular vesicles on OA. Third, these findings hold promise for clinical application, suggesting that HUCMSC-derived EVs are a potential treatment approach for OA management.

The limitations of this study include the following. First, although the mouse model provided valuable insights, extrapolating the findings to human patients with OA may require further validation in clinical studies. Second, this study primarily focused on short-term outcomes; the long-term effects of HUCMSC-derived extracellular vesicle therapy on OA progression remain unclear. Third, although this study demonstrated efficacy in a preclinical setting, clinical trials are required to evaluate the safety and effectiveness of HUCMSC-derived extracellular vesicle therapy in patients with OA. Fourth, the relationship between miR-7704 and MMP13 needs a more comprehensive experimental design. Another limitation of this study is the use of ExoQuick for EV isolation, as its inclusion of polyethylene glycol and lack of sterilization may have introduced contaminants, potentially limiting the application of isolated EVs in functional in vitro or in vivo studies.

## 5. Conclusions

The findings from the presented figures collectively demonstrate that EVs derived from HUCMSCs significantly improved the health and joint function of OA mice. Improvements were observed in Rotarod performance, cartilage histology, and key markers of cartilage integrity, such as type II collagen and aggrecan, highlighting the restorative potential of HUCMSC-derived exosomes on damaged cartilage. While a reduction in MMP-13 expression was noted, particularly in the lenti-miR7704-e-treated group, the overall therapeutic effects of HUCMSC exosomes appear to stem from a broader, multifaceted mechanism rather than exclusively from MMP-13 suppression. These results suggest that EVs from HUCMSCs, regardless of genetic modifications, are effective in mitigating OA pathology and improving joint health, positioning them as a promising therapeutic option for OA treatment.

## Figures and Tables

**Figure 1 cells-14-00082-f001:**
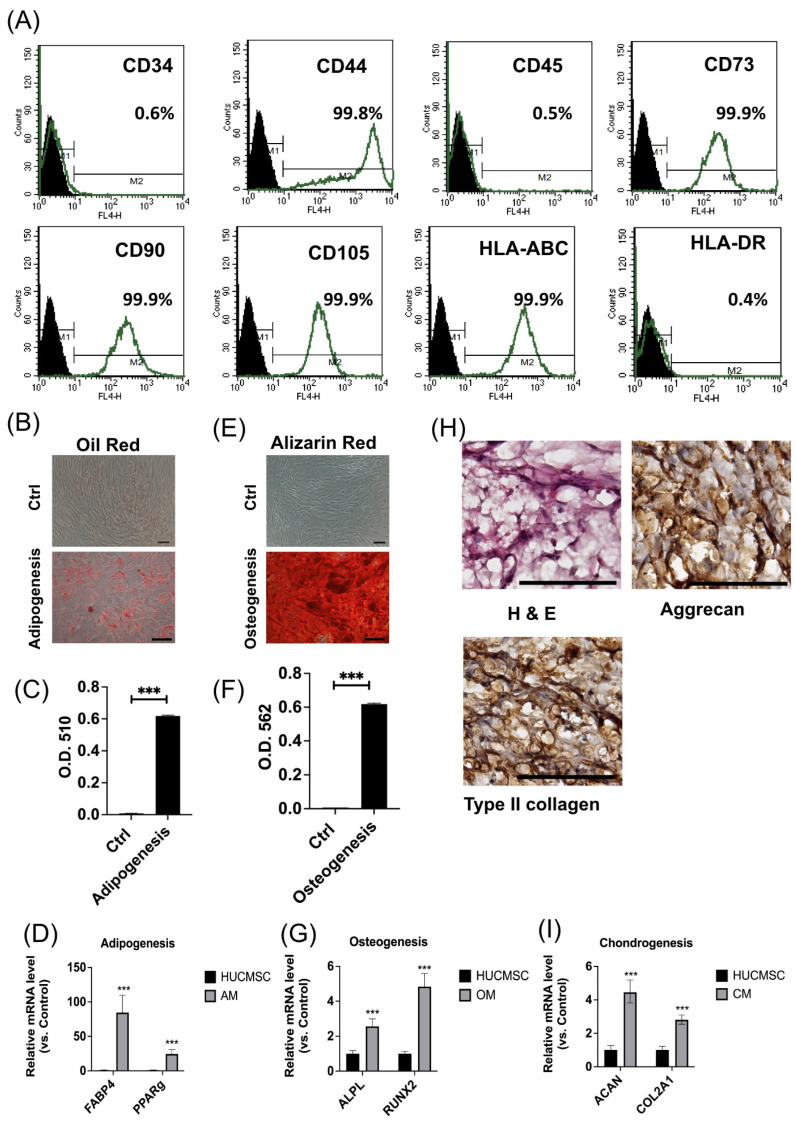
Characteristics of HUCMSCs. (**A**) Surface marker expression of HUCMSCs. (**B**–**D**) Adipogenesis. (**B**) Oil Red O staining of adipocytes differentiated from HUCMSCs. (**C**) Quantification of Oil Red staining at optical density (OD) 510. (**D**) Gene expressions of adipocyte-related genes (*FABP4* and *PPAR-γ*). (**E**–**G**) Osteogenesis. (**E**) Alizarin Red staining of HUCMSC-differentiated osteoblasts. (**C**) Quantification of Alizarin Red staining at OD 562. (**D**) Gene expressions of osteoblast-related genes (*APAL* and *RUNX2*). (**H**) Hematoxylin and eosin staining; immunohistochemistry of type II collagen and aggrecan of HUCMSC-differentiated chondrocytes. (**I**) Gene expressions of chondrocyte-related genes (*ACAN* and *COL2A1*). Scale bar = 100 μm. *** *p* < 0.001.

**Figure 2 cells-14-00082-f002:**
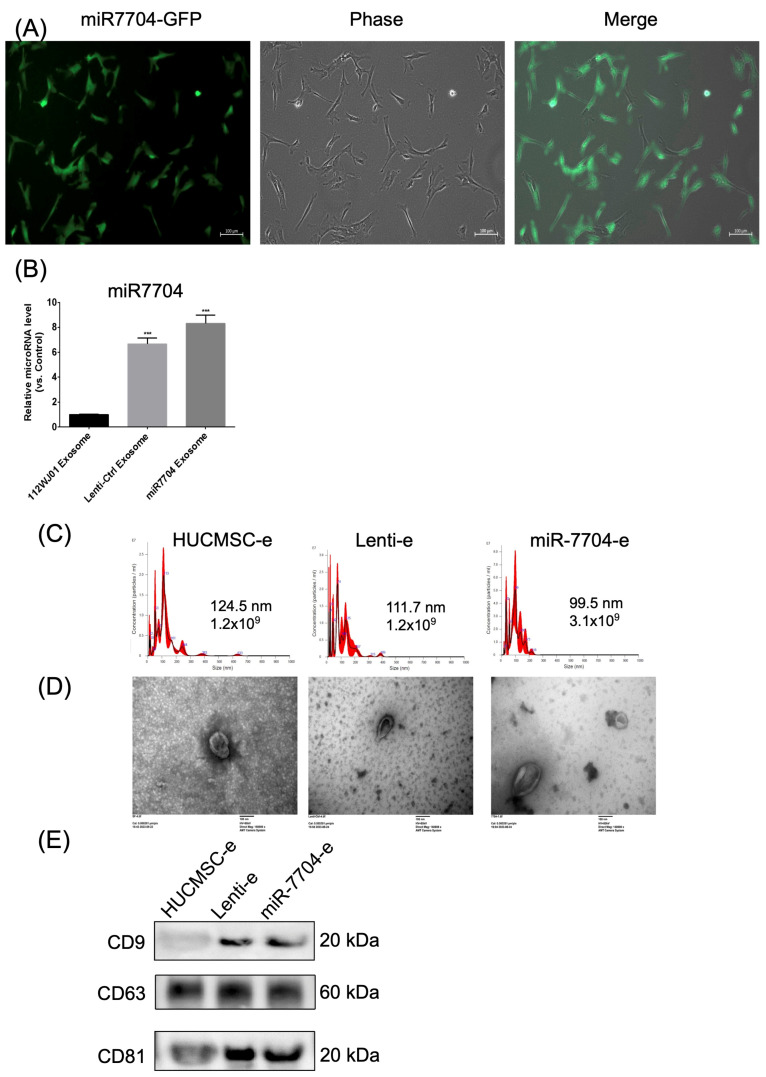
Characteristics of extracellular vesicles (EVs) derived from miR-7704-transfected HUCMSCs. (**A**) HUCMSCs transfected with GFP-tagged miR7704. GFP is revealed to be successfully transfected into HUCMSCs. (**B**) qPCR shows the gene expression of miR-7704 in the EVs derived from HUCMSC-, lenti-ctrl-, and miR-7704-transfected HUCMSCs. (**C**) Nanoparticle tracking analysis for measuring the size and concentration of EVs. (**D**) Transmission electron microscopy is used to observe the EVs; scale bar = 100 nm. (**E**) Western blot characterization of EVs’ marker proteins (CD9, 63, and 81). *** *p* < 0.001. Original images of (**E**) can be found in Appendix A.

**Figure 3 cells-14-00082-f003:**
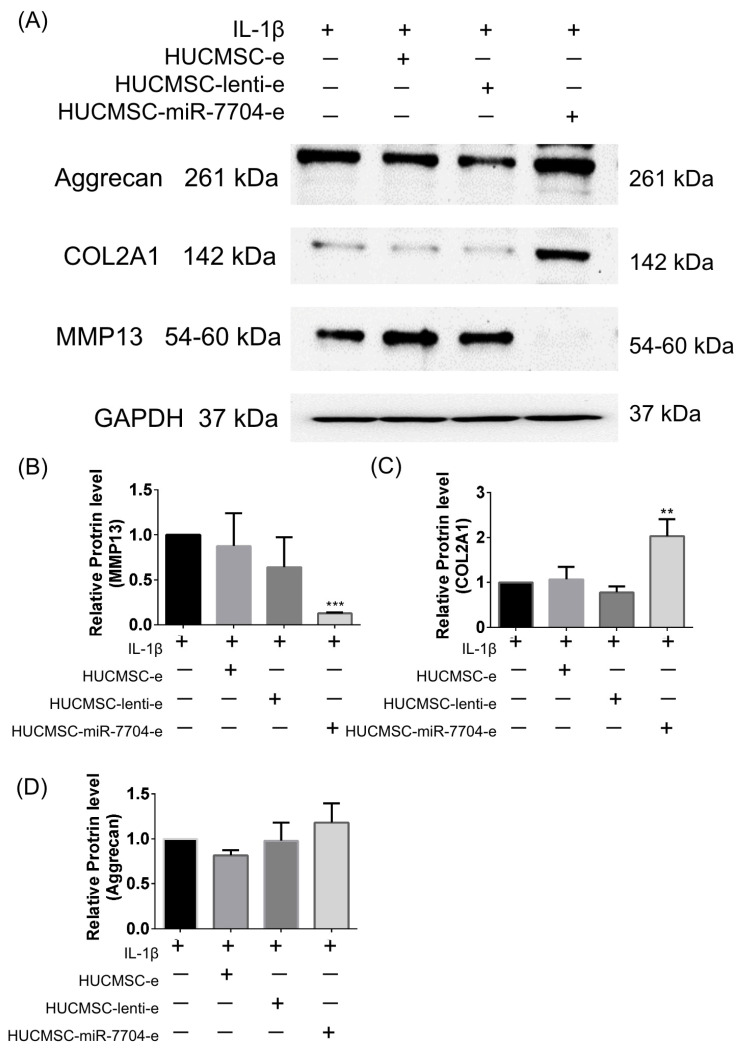
Western blot analysis of chondrocytes treated with IL-1beta 10ng/mL for 24 h, and then various extracellular vesicles (EVs) are treated. (**A**) Representative image of Western blotting. (**B**) Quantification of MMP13 (*n* = 3). (**C**) Quantification of type II collagen (COL2A1) (*n* = 3). (**D**) Quantification of aggrecan (*n* = 3). ** *p* < 0.01, *** *p* < 0.001. HUCMSCs: human umbilical cord mesenchymal stem cells; e: EVs; miR: microRNA. Original images of (**A**) can be found in Appendix A.

**Figure 4 cells-14-00082-f004:**
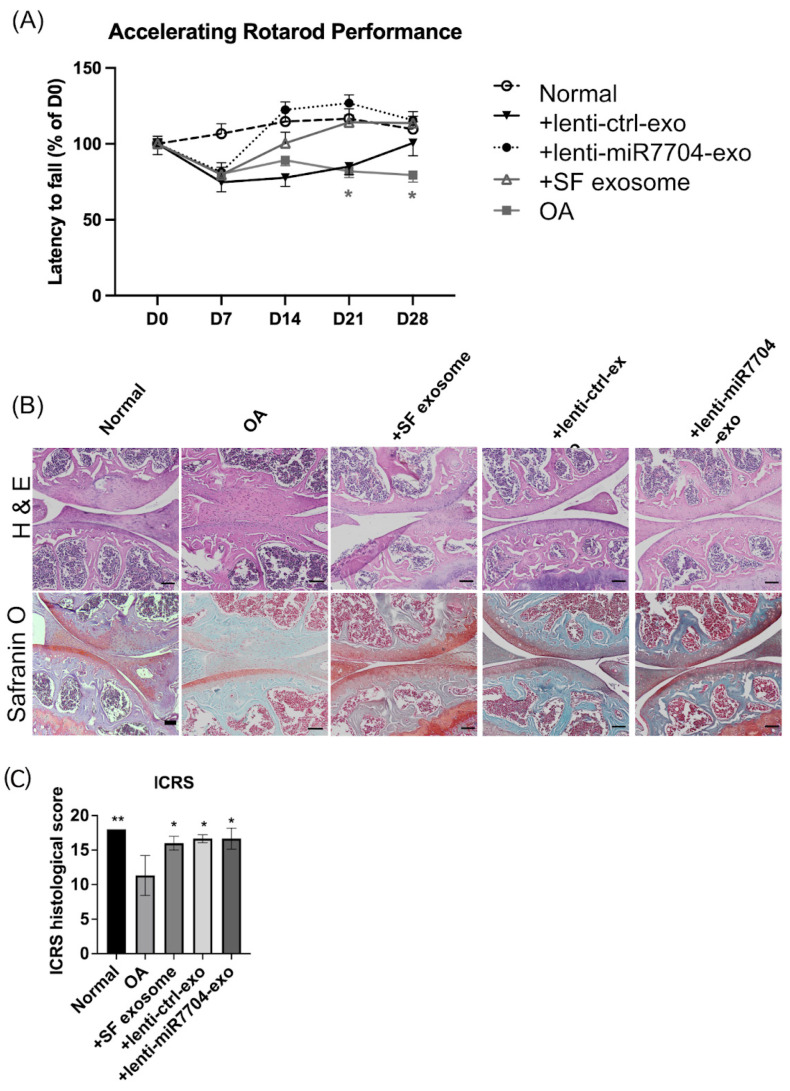
Rotarod performance test, histology, and The International Cartilage Repair Society (ICRS) score of osteoarthritis (OA) mice treated with HUCMSCs with or without transfection with lenti-ctrl or lenti-miR-7704 (*n* = 3 in each group). (**A**) Compared to the OA group, the OA mice that received HUCMSCs-EVs and lenti-miR-7704-EVs showed significant improvement after day 21. OA transplanted with lenti-ctrl EVs showed improvement on day 28. Ctrl: control, * *p* < 0.05. (**B**) The histology of the mouse OA model after 28 days of experiments (hematoxylin and eosin and Safranin O staining). The representative image of each group is illustrated. Scale bar = 100 μm. (**C**) The ICRS scores of the cartilage after various treatments. OA: osteoarthritis. A statistical test was performed using one-way ANOVA with a post hoc and Bonferroni test. * *p* < 0.05, ** *p* < 0.01.

**Figure 5 cells-14-00082-f005:**
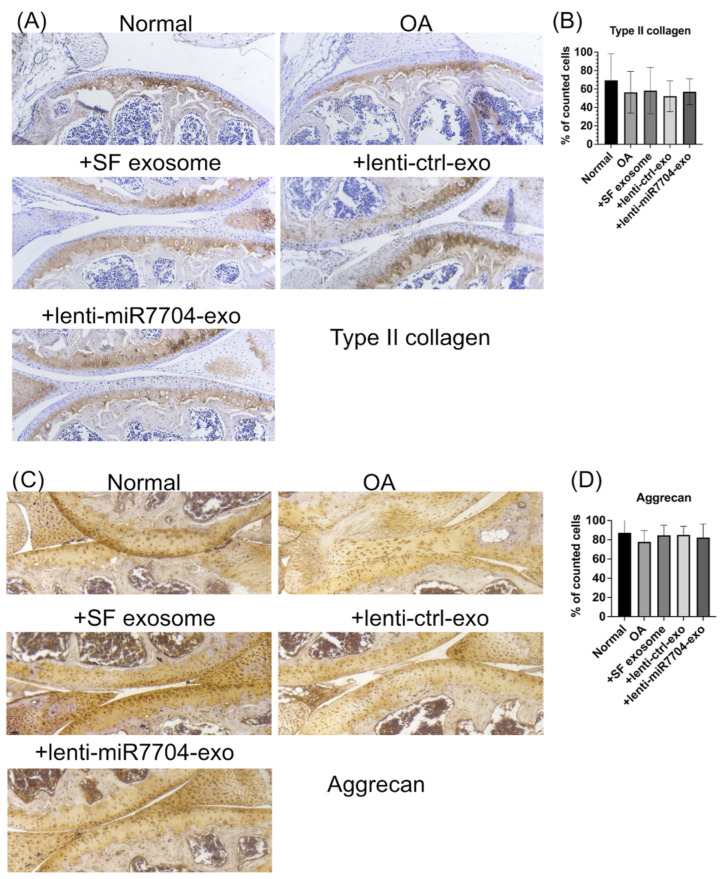
Immunohistochemical analysis of cartilage in mouse osteoarthritis (OA) model. (**A**) Type II collagen staining image from normal knee, OA, HUCMSCs-EVs, lenti-control-EVs, and miR-7704-EVs. (**B**) Quantitative analysis of the percentage of positively stained cells across five randomly selected fields (mean ± standard deviation). (**C**) Representative image of aggrecan for each group. (**D**) Quantitative analysis of the percentage of positively stained cells across five randomly selected fields (mean ± standard deviation).

**Figure 6 cells-14-00082-f006:**
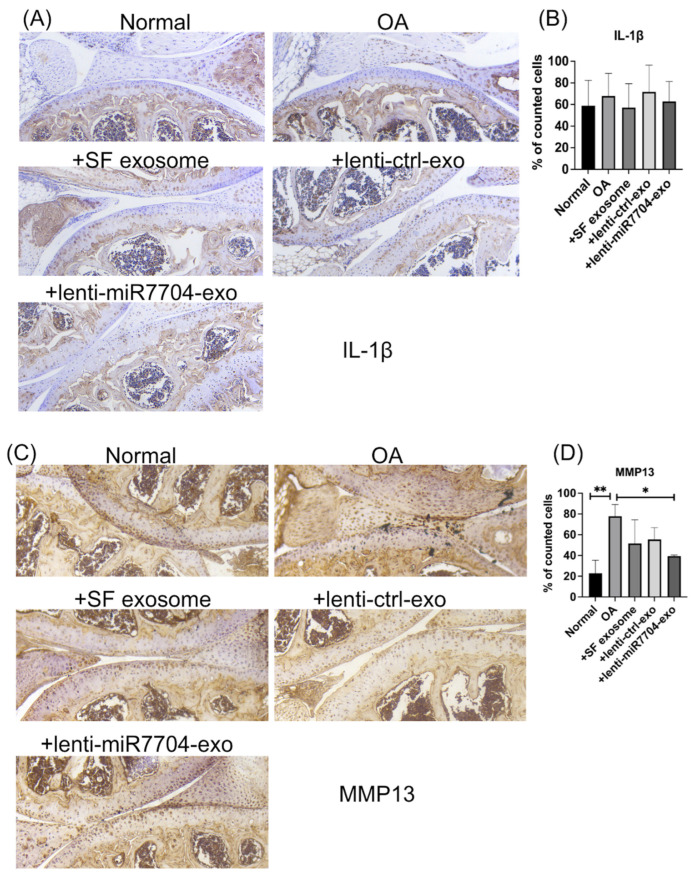
Immunohistochemical analysis of cartilage in mouse osteoarthritis (OA) model. (**A**) IL-1β staining image from normal knee, OA, HUCMSCs-EVs, lenti-control-EVs, and lenti-miR-7704-EVs. (**B**) Quantitative analysis of the percentage of positively stained cells across five randomly selected fields (mean ± standard deviation). (**C**) Representative image of MMP13 for each group. (**D**) Quantitative analysis of the percentage of positively stained cells across five randomly selected fields (mean ± standard deviation). * *p* < 0.05, ** *p* < 0.01.

**Table 1 cells-14-00082-t001:** Primer sequence of candidate genes.

Gene Name	Forward Sequence (5′-3′)	Reverse Sequence (5′-3′)	Product Size (bp)
*PPARγ*	AGCCTCATGAAGAGCCTTCCA	TCCGGAAGAAACCCTTGCA	120
*FABP4*	ATGGGATGGAAAATCAACCA	GTGGAAGTGACGCCTTTCAT	87
*ALPL*	CCACGTCTTCACATTTGGTG	GCAGTGAAGGGCTTCTTGTC	96
*RUNX2*	CGGAATGCCTCTGCTGTTAT	TTCCCGAGGTCCATCTACTG	174
*ACAN*	GAGATGGAG GGTGAGGTC	ACGCTGCCTCGGGCTTC	443
*COL2A1*	GGACTTTTCTCCCCTCTCT	GACCCGAAGGTCTTACAGGA	104
*GAPDH*	GAAGGTGAAGGTCGGAGTC	GAAGATGGTGATGGGATTTC	172

## Data Availability

The datasets generated and/or analyzed in the current study are available from the corresponding author upon reasonable request.

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
