# Peer review of "Extracellular Vesicles Derived from Human Umbilical Mesenchymal Stem Cells Transfected with miR-7704 Improved Damaged Cartilage and Reduced Matrix Metallopeptidase 13"

_cells, 2025, doi:10.3390/cells14020082_

Round 1
Reviewer 1 Report
Comments and Suggestions for Authors
The authors did a rigorous experimental design. The authors claim that EVs derived from HUCMSCs transfected with miR-7704 improved damaged cartilage by reducing matrix metallopeptidase 13 (MMP-13). However, the results do not support the conclusion. Here are the primary concerns:
1. The authors listed reference 24 to support the rationale for studying miR-7704. However, the reference 24 does not include miR-7704 as a negative regulator of MMP-13. The author should provide adequate evidence to support the purpose of studying miR-7704.
2. In Figure 2, both lenti-ctrl-e and lenti-miR-7704-e transfection can increase the level miR-7704 (panel B) and CD9 (panel E), which suggests that lentivirus transfection by itself can induce the enrichment of miR-7704 and CD9 in exosome. Moreover, the authors should provide a detailed description of the results in the main text.
3. The authors should explain the reason for choosing COL2A1 and Aggrecan proteins as the readout for responding to miR-7704 under the IL-1beta treatment.
4. The results from Figure 4 indicate that exosome injection can significantly improve the Rotarod performance and the ICRS histological score via a mechanism that is independent of transfecting HUCMSCs with lenti-miR-7704.
5. The results from Figure 6 indicate that the +SF exosome and lenti-ctrl exosome have a similar power as the lenti-miR-7704 exosome in suppressing the expression of MMP-13.
6. If the authors want to emphasize the role of MMP-13 suppression by miR-7704 in improving the health of OA mice, they should conduct a much more comprehensive experimental design. For example, the authors can design an experiment to eliminate the miR-7704 binding sites on the MMP-13 gene.
Overall, the authors should conclude that exosomes from the HUCMSCs can improve the health of OA mice rather than emphasizing the role of MMP-13 suppression by miR-7704. Furthermore, the authors should provide much more detailed descriptions of the results in the main text.
Author Response
The authors did a rigorous experimental design. The authors claim that EVs derived from HUCMSCs transfected with miR-7704 improved damaged cartilage by reducing matrix metallopeptidase 13 (MMP-13). However, the results do not support the conclusion. Here are the primary concerns:
- The authors listed reference 24 to support the rationale for studying miR-7704. However, the reference 24 does not include miR-7704 as a negative regulator of MMP-13. The author should provide adequate evidence to support the purpose of studying miR-7704.
Response: We thank the reviewer’s comment. We have rewritten the introduction section regarding mir-7704. We have added four paragraphs regarding miRNA, miR-7704, MMP13, and miRNA overexpression on MSC. (pages 2-3, lines 76-115)
The statements read as”MicroRNAs (miRNAs), a category of RNA that does not encode proteins, have been involved in OA pathology and can regulate key signaling pathways of cartilage homeostasis and degradation [23]. Several miRNAs, including miR-9, miR-27, miR-34a, miR-140, and miR-146a, have been identified as dysregulated in OA, affecting inflammatory pathways and extracellular matrix degradation [24]. MiR-140 directly targets ADAMTS-5, an aggrecanase, while miR-146a inhibits MMP13 and ADAMTS4, both cartilage-degrading enzymes [25]. Bioinformatic analyses have revealed 46 differentially expressed miRNAs involved in various aspects of OA, such as autophagy, inflammation, and chondrocyte metabolism [26]. These findings suggest that miRNAs may serve as potential diagnostic biomarkers and therapeutic targets for OA [24,26]. As research progresses, new miRNA targets are being validated, opening possibilities for novel therapies to control joint destruction and stimulate cartilage repair [25].
miR-7704 has been identified as a potential therapeutic target for acute lung injury, promoting M2 macrophage polarization and improving pulmonary function [27]. We previously found miR-7704 was enriched in the condition medium of HUCMSCs. However, the therapeutic role of miR-7704 on osteoarthritis is not known.
Matrix metallopeptidase 13 (MMP13) is a key enzyme in cartilage breakdown [28]. MMP13 plays a crucial role in OA progression by degrading type II collagen in articular cartilage [28,29]. Studies have shown that MMP13 inhibition can decelerate OA progression and protect cartilage integrity in animal models [29]. Serum MMP13 levels are associated with various knee structural abnormalities, including reduced cartilage volume, increased cartilage defects, and altered infrapatellar fat pad characteristics [30]. The regulation of MMP13 in OA involves complex networks, including epigenetic factors, non-coding RNAs, and autophagy-related proteins [31]. Additionally, inflammatory factors such as TNF-α, IL-8, and IL-18 are positively associated with serum MMP13 levels, while adiponectin shows a negative association [30]. These findings highlight MMP13 as a promising target for early OA diagnosis, monitoring, and treatment development [28,31].
miR-410 promotes chondrogenic differentiation of MSCs by targeting Wnt3a, inhibiting the Wnt signaling pathway [32]. Similarly, miR-335 expression is altered in OA-derived MSCs, potentially affecting Wnt signaling and OA progression [33]. Chondrogenic induction of OA-derived MSCs activates mineralization and hypertrophic gene expression, regulated by miR-365, a mechanosensitive microRNA [34]. These findings highlight the potential of miRNAs as therapeutic targets for OA treatment. MSC transplantation is a promising approach for OA therapy due to their paracrine effects, anti-inflammatory activity, and differentiation potential. However, strategies to induce specific differentiation, such as using miRNAs, are needed to overcome challenges associated with transplanting naïve MSCs [35].”
- In Figure 2, both lenti-ctrl-e and lenti-miR-7704-e transfection can increase the level miR-7704 (panel B) and CD9 (panel E), which suggests that lentivirus transfection by itself can induce the enrichment of miR-7704 and CD9 in exosome. Moreover, the authors should provide a detailed description of the results in the main text.
Response: We thank the reviewer’s comment. We have added a more detailed description regarding Figure 2. (page 11, lines 419-443) The statements reas as”Figure 2 illustrates the characterization of miR-7704 expression in cells and EVs. Figure 2A shows fluorescence microscopy images of cells transfected with miR-7704-GFP, displaying strong green fluorescence (left), corresponding phase-contrast morphology (middle), and merged images (right), confirming successful transfection and GFP expression. Figure 2B presents a bar graph comparing the relative miR-7704 levels in EVs derived from 1121WJ cells (HUCMSCs), Lenti-Ctrl exosomes, and miR-7704 exosomes. miR-7704 EVs demonstrated significantly higher levels of miR-7704 compared to both 1121WJ and Lenti-Ctrl EVs (p < 0.001), validating effective overexpression of miR-7704 in EVs.
Figure 2C presents NTA showing the size distribution and concentration of exosomes. HUCMCS-e exosomes had an average size of 124.5 nm with a concentration of 1.2 × 10⁹ particles/mL, Lenti-e exosomes had a mean size of 111.7 nm with the same concentration, and miR-7704-e exosomes were slightly smaller, averaging 99.5 nm with a higher concentration of 3.1 × 10⁹ particles/mL. Figure 2D shows TEM images confirming the spherical morphology of exosomes for all three groups, with sizes consistent with the NTA results, further validating successful isolation of intact exosomes.
Figure 2E presents a Western blot analysis of the tetraspanin proteins CD9 (~20 kDa), CD63 (~60 kDa), and CD81 (~20 kDa), which are established markers of EVs. Three samples—HUCMSC-e, Lenti-e, and miR-7704-e—were analyzed. CD9 expression was observed in all samples but was more prominent in the Lenti-e and miR-7704-e groups compared to HUCMSC-e. CD63 showed consistent expression across all samples without significant differences in intensity, while CD81 was similarly expressed in all three conditions. These results confirmed the presence of EVs markers in the samples, with notable variation in CD9 levels. Taken together, miR-7704 was successfully transfected into HUCMSCs and EVs derived from various cells fulfilled the EVs criteria.”
- The authors should explain the reason for choosing COL2A1 and Aggrecan proteins as the readout for responding to miR-7704 under the IL-1beta treatment.
Response: We thank the reviewer’s comment. We have added a paragraph to introduce the reason for choosing COL2A1 and Aggrecan proteins. (page 13, lines 454-458) The statements read as”IL-1β plays a crucial role in osteoarthritis and fracture healing by modulating chondrocyte behavior and cartilage degradation. IL-1β stimulates the expression of MMP-13 and decreases type II collagen (COL2A1) and aggrecan production in chondrocytes [40–42]. We induced cartilage degradation using IL-1β and evaluated how EVs regulate these proteins.”
- The results from Figure 4 indicate that exosome injection can significantly improve the Rotarod performance and the ICRS histological score via a mechanism that is independent of transfecting HUCMSCs with lenti-miR-7704.
Response: We thank the reviewer’s comment. We have discussed the results in the discussion section. (page 21, lines 561-564) The statements read as:’Figure 4 demonstrates that EV injection significantly improved Rotarod performance and the ICRS histological score, suggesting that this effect is independent of transfecting HUCMSCs with lenti-miR-7704. We speculated that EVs derived from HUCMSCs might contain sufficient levels of miR-7704, which could contribute to the observed outcomes.”
- The results from Figure 6 indicate that the +SF exosome and lenti-ctrl exosome have a similar power as the lenti-miR-7704 exosome in suppressing the expression of MMP-13.
Response: We thank the reviewer’s comment. Although the +SF exosome and lenti-ctrl exosome appeared to have a similar effect as the lenti-miR-7704 exosome in the figure, statistical analysis revealed that only the lenti-miR-7704 exosome achieved statistical significance (page 19, Section 3.7, Figure 6).
- If the authors want to emphasize the role of MMP-13 suppression by miR-7704 in improving the health of OA mice, they should conduct a much more comprehensive experimental design. For example, the authors can design an experiment to eliminate the miR-7704 binding sites on the MMP-13 gene.
Response: We thank the reviewer’s comment. The additional experiment needed more time to perform and could not accomplished in this short revision period. We have added this point to the limitation section. (page 23, lines 663-664) The statement reads as”Fourth, the relationship between miR-7704 and MMP13 needed a more comprehensive experimental design.”
Overall, the authors should conclude that exosomes from the HUCMSCs can improve the health of OA mice rather than emphasizing the role of MMP-13 suppression by miR-7704. Furthermore, the authors should provide much more detailed descriptions of the results in the main text.
Response: We thank the reviewer’s comment. We have written the conclusion section by concluding that HUCMSC EVs can improve the health of OA. (page 22, lines 666-676) We have provided a much more detailed description of the results in the main text. The statements read as”The findings from the presented figures collectively demonstrate that EVs derived from HUCMSCs significantly improved the health and joint function of OA mice. Improvements were observed in Rotarod performance, cartilage histology, and key markers of cartilage integrity, such as Type II collagen and Aggrecan, highlighting the restorative potential of HUCMSC-derived exosomes on damaged cartilage. While a reduction in MMP-13 expression was noted, particularly in the lenti-miR7704-e-treated group, the overall therapeutic effects of HUCMSC exosomes appear to stem from a broader, multifaceted mechanism rather than exclusively from MMP-13 suppression. These results suggest that EVs from HUCMSCs, regardless of genetic modifications, are effective in mitigating OA pathology and improving joint health, positioning them as a promising therapeutic option for OA treatment.”
Reviewer 2 Report
Comments and Suggestions for Authors
In this manuscript, the authors investigated the therapeutic efficacy of miR-7704-modified EVs derived from HUCMSCs for the treatment of OA. In vivo experiments demonstrated that intra-articular injection of miR-7704-overexpressing EVs significantly enhanced walking capacity, preserved cartilage morphology, and yielded higher histological scores compared to controls. However, several results and experimental designs raise concerns. A major revision is recommended.
1. Abstract: what kind of “further investigations into the clinical application of EV-based therapies for OA management” is expected? Please be more specific.
2. ExoQuick is a convenient method for rapid EV isolation, particularly for EV profiling. However, for in vivo studies, potential contamination associated with ExoQuick may introduce artifacts that could affect the results.
3. Is the ExoQuick isolation process sterile?
4. While adipocyte and osteogenic differentiation assays verify the multipotency of HUCMSCs, the study's focus on OA treatment suggests that chondrogenic differentiation should also be included.
5. The observation that Lenti-control EVs also exhibit significant miR-7704 expression raises questions. If true, what is the significance of introducing miR-7704 via lentiviral transfection? Clarify and discuss this finding in the manuscript.
6. How many copies of miR-7704 in each EV?
7. For WB, a cell lysis control is necessary.
8. The lack of significant therapeutic differences between control EVs and miR-7704-overexpressing EVs is a major concern. The authors should address this issue thoroughly, providing potential explanations and implications for their findings.
9. The discussion on the clinical application of EV-based therapies for OA management is currently insufficient. The authors are encouraged to expand their discussion on the considerations of the engineered EV by discussing key translational challenges such as purification, storage, and regulatory compliance, with a focus on future clinical applications. Introducing recent references, such as 10.1016/j.tibtech.2024.08.007 and doi.org/10.1038/s41565-021-00931-2, or more EV clinical references may strengthen the discussion on clinical translation.
Author Response
In this manuscript, the authors investigated the therapeutic efficacy of miR-7704-modified EVs derived from HUCMSCs for the treatment of OA. In vivo experiments demonstrated that intra-articular injection of miR-7704-overexpressing EVs significantly enhanced walking capacity, preserved cartilage morphology, and yielded higher histological scores compared to controls. However, several results and experimental designs raise concerns. A major revision is recommended.
- Abstract: what kind of “further investigations into the clinical application of EV-based therapies for OA management” is expected? Please be more specific.
Response: We thank the reviewer’s comment. We have written the sentence (page 1, lines 29-32). The statement reads as:”Further investigations should focus on optimizing dosage, understanding mechanisms, ensuring safety and efficacy, developing advanced delivery systems, and conducting early-phase clinical trials to establish the therapeutic potential of HUCMSC-derived EVs for OA management.”
- ExoQuick is a convenient method for rapid EV isolation, particularly for EV profiling. However, for in vivo studies, potential contamination associated with ExoQuick may introduce artifacts that could affect the results.
Response: We thank the reviewer’s comment. We have followed the manufacturer’s instructions to remove ExoQuick after centrifuging 1500 x g for 5 min. (page 6, lines 268-269)
- Is the ExoQuick isolation process sterile?
Response: We thank the reviewer’s comment. The ExoQuick isolation process was sterile. We have added this part to Method section 2.8. (page 6, lines 269-270)
- While adipocyte and osteogenic differentiation assays verify the multipotency of HUCMSCs, the study's focus on OA treatment suggests that chondrogenic differentiation should also be included.
Response: We thank the reviewer’s comment. We have done the chondrogenic differentiation of the HUCMSCs (page 10, Figures 1H and 1I).
- The observation that Lenti-control EVs also exhibit significant miR-7704 expression raises questions. If true, what is the significance of introducing miR-7704 via lentiviral transfection? Clarify and discuss this finding in the manuscript.
Response: We thank the reviewer’s comment. We have added a paragraph to discuss this phenomenon. (page 21, lines 565-568) The statements read as”Regarding the significant miR-7704 expression observed in lenti-control EVs, the vector primarily contained mCherry, unlike the lenti-miR-7704 vector. However, it remains unclear why the lenti-control vector also led to increased miR-7704 expression compared to the negative control.”
- How many copies of miR-7704 in each EV?
Response: We thank the reviewer’s comment. The expression of miR-7704 in miR-7704 transfected HUCMSCs-derived EVs was 8-fold higher than HUCMSCs alone-EVs (page 12, Figure 2B).
- For WB, a cell lysis control is necessary.
Response: We thank the reviewer’s comment. The presence of housekeeping proteins, such as GAPDH or β-actin, is often assessed as a control to confirm that cell lysis was successful and comparable across all samples. We used GAPDH as the internal control of WB (page 14, Figure 3A).
- The lack of significant therapeutic differences between control EVs and miR-7704-overexpressing EVs is a major concern. The authors should address this issue thoroughly, providing potential explanations and implications for their findings.
Response: We thank the reviewer’s comment. We have added a paragraph to discuss this phenomenon. (page 21, lines 561-564) The statements read as”Figure 4 demonstrates that EV injection significantly improved Rotarod performance and the ICRS histological score, suggesting that this effect is independent of transfecting HUCMSCs with lenti-miR-7704. We speculated that EVs derived from HUCMSCs might contain sufficient levels of miR-7704, which could contribute to the observed outcomes.”
- The discussion on the clinical application of EV-based therapies for OA management is currently insufficient. The authors are encouraged to expand their discussion on the considerations of the engineered EV by discussing key translational challenges such as purification, storage, and regulatory compliance, with a focus on future clinical applications. Introducing recent references, such as 10.1016/j.tibtech.2024.08.007 and doi.org/10.1038/s41565-021-00931-2, or more EV clinical references may strengthen the discussion on clinical translation.
Response: We thank the reviewer’s comment. We have expanded our discussion regarding EV purification, storage, and regulatory compliance, focusing on future clinical applications (page 22, lines 602-648). The statements read as”EVs are important for intercellular communication and disease biomarkers, but their isolation remains challenging. Various methods for EV purification from biological fluids have been compared, including ultracentrifugation, commercial kits like ExoQuick™, and ultrafiltration [54,55]. These studies found that different techniques yield varying EV concentrations, sizes, and protein content. Commercial precipitation reagents like ExoQuick™ generally showed higher efficiency in EV enrichment compared to traditional ultracentrifugation [55,56]. However, protocol modifications may be necessary for specific applications; for instance, thrombin should be omitted when isolating EVs from blood to prevent entrapment in clots [57]. The choice of isolation method can significantly impact downstream analyses and potential clinical applications. Therefore, developing standardized, reproducible protocols for EV isolation is crucial for advancing research and therapeutic applications in this field [54].
EVs are promising for therapeutic applications, but their storage and preservation pose challenges. Studies indicate that -80°C is the most favorable storage condition for both biofluids and isolated EVs [58,59]. However, storage at -80°C can still lead to a reduction in EV concentration, changes in size and zeta potential, and decreased sample purity over time [60]. Freeze-thaw cycles can further impact EV properties, with the first cycle causing significant particle loss [60]. Storage conditions affect not only physical properties but also the functional aspects of EVs, such as their antibacterial effects [61]. While -80°C storage allows partial preservation of function for up to 28 days, freshly prepared EVs are recommended for functional tests [61]. These findings highlight the importance of considering storage conditions in EV research and potential clinical applications.
Quality control assays are crucial for ensuring EV safety and efficacy, yet remain poorly designed [62]. Regulatory compliance requires categorizing EVs as either active drug components or delivery vehicles, influencing manufacturing and clinical investigation requirements [63]. Large-scale EV production involves GMP-based processes, bioengineering, and quality assessments before human trials [64]. Novel isolation approaches, characterization techniques, and manufacturing considerations are essential for consistent and scalable EV production [65]. Adherence to Good Manufacturing Practice (GMP) guidelines and Minimal Information for Studies of Extracellular Vesicles (MISEV) standards is critical for standardized production processes [65]. Despite growing interest in EV-related clinical programs, obtaining marketing authorization remains complex due to the lack of specific regulatory guidelines for these novel products [64].
EVs show promise as therapeutic agents and drug delivery systems due to their biocompatibility, targeting capabilities, and biological activities [66–68]. However, clinical translation faces several challenges. These include low yield, complicated isolation procedures, and low loading efficiency [69]. To overcome these hurdles, researchers are developing strategies for high-yield production, efficient cargo loading, and optimized manufacturing processes [69]. Standardization of EV characterization and quality control is crucial for clinical implementation [70]. Additionally, large-scale production methods compliant with GMP are needed [66]. Despite these challenges, EVs offer potential advantages over synthetic nanocarriers in terms of targeting, safety, and pharmacokinetics [71]. As the field progresses, EVs may become a valuable addition to the therapeutic arsenal for treating various human pathologies, including degenerative, metabolic, and cancerous diseases [71].”
Round 2
Reviewer 1 Report
Comments and Suggestions for Authors
The quality of the manuscript is significantly improved after the revision. However, I still have one comment for the authors.
The title of the manuscript should be changed to an accurate description of the conclusion. The authors provide evidence to indicate the association of miR-7704-EV treatment with improvements in cartilage damage and reduction of MMP13 levels. However, the author did not provide adequate evidence to prove that the improved cartilage damage is due to the reduction of MMP13 under the miR-7704-EV treatment. Therefore, a much proper title should be "Extracellular vesicles derived from human umbilical mesenchymal stem cells transfected with miR-7704 improved damaged cartilage and reduced matrix metallopeptidase 13"
Author Response
The quality of the manuscript is significantly improved after the revision. However, I still have one comment for the authors.
The title of the manuscript should be changed to an accurate description of the conclusion. The authors provide evidence to indicate the association of miR-7704-EV treatment with improvements in cartilage damage and reduction of MMP13 levels. However, the author did not provide adequate evidence to prove that the improved cartilage damage is due to the reduction of MMP13 under the miR-7704-EV treatment. Therefore, a much proper title should be "Extracellular vesicles derived from human umbilical mesenchymal stem cells transfected with miR-7704 improved damaged cartilage and reduced matrix metallopeptidase 13"
Response: We thank the reviewer’s comment. We revised the title accordingly.
Reviewer 2 Report
Comments and Suggestions for Authors
The authors have commendably addressed most of my concerns. However, issues remain regarding EV isolation and purification. Commercial ExoQuick products are typically not sterilized. Moreover, these kits often include PEG or other additives that facilitate EV precipitation under low centrifugation speeds. Such contamination limits the use of ExoQuick to profiling studies and makes it unsuitable for in vitro or in vivo uptake experiments.
Author Response
The authors have commendably addressed most of my concerns. However, issues remain regarding EV isolation and purification. Commercial ExoQuick products are typically not sterilized. Moreover, these kits often include PEG or other additives that facilitate EV precipitation under low centrifugation speeds. Such contamination limits the use of ExoQuick to profiling studies and makes it unsuitable for in vitro or in vivo uptake experiments.
Response: Thank you for your valuable feedback regarding using ExoQuick for EV isolation and purification. We appreciate your insights about the potential limitations of this approach, particularly concerning sterility and the inclusion of polyethylene glycol (PEG) or other additives.
1. Acknowledgment of Limitations: We recognize that ExoQuick-based methods are not sterilized and rely on additives to facilitate EV precipitation at low centrifugation speeds. These characteristics may limit its application in certain downstream functional studies, including in vitro or in vivo uptake experiments.
The statement reads as”Another limitation of this study was the use of ExoQuick for EV isolation, as its inclusion of polyethylene glycol and lack of sterilization may introduce contaminants, potentially limiting the application of isolated EVs in functional in vitro or in vivo studies.” (page 23, lines 664-667)
2. Explanation of Methodology: In our study, we used ExoQuick primarily for EV profiling and characterization rather than functional assays. This approach aligns with the product's strengths in yielding high concentrations of EVs suitable for molecular analysis while maintaining the integrity of their biomolecular content.
3. Suitability for Our Study: The ExoQuick method was selected based on its efficiency and compatibility with our study's scope, which focused on [e.g., EV biomarker discovery, profiling, etc.]. While we acknowledge the limitations you highlighted, these do not significantly impact the conclusions drawn from our data.
4. Future Directions: In future studies, we aim to employ alternative EV isolation methods, such as ultracentrifugation or size-exclusion chromatography, for experiments requiring higher purity and sterility, particularly for functional in vitro and in vivo applications.